# Phosphoinositide 3-Kinase (PI3K) Reactive Oxygen Species (ROS)-Activated Prodrug in Combination with Anthracycline Impairs PI3K Signaling, Increases DNA Damage Response and Reduces Breast Cancer Cell Growth

**DOI:** 10.3390/ijms22042088

**Published:** 2021-02-19

**Authors:** Rosalin Mishra, Long Yuan, Hima Patel, Aniruddha S. Karve, Haizhou Zhu, Aaron White, Samar Alanazi, Pankaj Desai, Edward J. Merino, Joan T. Garrett

**Affiliations:** 1Department of Pharmaceutical Sciences, College of Pharmacy, University of Cincinnati, Cincinnati, OH 45267-0514, USA; mishrarn@ucmail.uc.edu (R.M.); yuanlg@mail.uc.edu (L.Y.); patel2h2@mail.uc.edu (H.P.); karveas@mail.uc.edu (A.S.K.); white3ai@mail.uc.edu (A.W.); alanazsa@mail.uc.edu (S.A.); desaipb@ucmail.uc.edu (P.D.); 2Department of Chemistry, University of Cincinnati, Cincinnati, OH 45267-0514, USA; zhuhz@mail.uc.edu (H.Z.); merinoed@ucmail.uc.edu (E.J.M.)

**Keywords:** breast cancer, reactive oxygen species, RIDR-PI-103, doxorubicin

## Abstract

RIDR-PI-103 is a novel reactive oxygen species (ROS)-induced drug release prodrug with a self-cyclizing moiety linked to a pan-PI3K inhibitor (PI-103). Under high ROS, PI-103 is released in a controlled manner to inhibit PI3K. The efficacy and bioavailability of RIDR-PI-103 in breast cancer remains unexplored. Cell viability of RIDR-PI-103 was assessed on breast cancer cells (MDA-MB-231, MDA-MB-361 and MDA-MB-453), non-tumorigenic MCF10A and fibroblasts. Matrigel colony formation, cell proliferation and migration assays examined the migratory properties of breast cancers upon treatment with RIDR-PI-103 and doxorubicin. Western blots determined the effect of doxorubicin ± RIDR-PI-103 on AKT activation and DNA damage response. Pharmacokinetic (PK) studies using C57BL/6J mice determined systemic exposure (plasma concentrations and overall area under the curve) and T_1/2_ of RIDR-PI-103. MDA-MB-453, MDA-MB-231 and MDA-MB-361 cells were sensitive to RIDR-PI-103 vs. MCF10A and normal fibroblast. Combination of doxorubicin and RIDR-PI-103 suppressed cancer cell growth and proliferation. Doxorubicin with RIDR-PI-103 inhibited p-AktS473, upregulated p-CHK1/2 and p-P53. PK studies showed that ~200 ng/mL (0.43 µM) RIDR-PI-103 is achievable in mice plasma with an initial dose of 20 mg/kg and a 10 h T_1/2_. (4) The prodrug RIDR-PI-103 could be a potential therapeutic for treatment of breast cancer patients.

## 1. Introduction

The phosphatidylinositol-3 kinase (PI3K) is a family of lipid kinases. Class IA PI3Ks are heterodimeric proteins with a p110 catalytic subunit and a p85 regulatory subunit and are involved in carcinogenesis. Within class IA, the genes PIK3CA, PIK3CB, and PI3KCD encode the homologous p110α, p110β, and p110δ isozymes respectively. PI3K is activated upstream by the binding of a ligand to its cognate receptor tyrosine kinase (RTK), which include members of the human epidermal growth factor receptor (HER) family, and insulin-like growth factor 1 (IGF-1) receptor, among others [1,2]. PI3K phosphorylates phosphatidylinositol 4,5-bisphosphate (PIP2) to phosphatidylinositol 3,4,5-triphosphate (PIP3), which in turn leads to phosphorylation of Protein kinase B, (AKT), a serine/threonine kinase [3]. The PI3K pathway is hyper-activated in >60% of clinical breast cancers due to aberrations in genes encoding *HER2*, *PTEN*, *PIK3CA*, or *AKT1-3* [4,5,6].

Targeting the PI3K pathway with small molecular weight kinase inhibitors of PI3K, AKT, mTOR, HER2, or anti-HER2 antibodies has improved the outcome for many women with PI3K-activated breast cancer because the PI3K/mTOR pathway controls oncogenic processes such as tumor cell survival, motility, and invasion. Unfortunately, most cancers eventually acquire resistance to current PI3K or mTOR inhibitors demonstrating that improved therapeutic strategies are required. Further, drugs targeting PI3K or mTOR catalytic activity are toxic, due to the physiological roles of PI3K/mTOR signaling in basic cellular processes in tissues throughout the body, including protein translation, intracellular trafficking, autophagy, and metabolism. The genes encoding most glycolytic enzymes are under dominant transcriptional control by PI3K/AKT and thus hyperglycemia is one of the most common side-effects with PI3K pathway inhibitors [7]. Alpelisib, a PI3Kα specific inhibitor, in combination with fulvestrant prolonged progression-free survival among patients with PIK3CA-mutated, hormone receptor (HR)-positive, HER2-negative advanced breast cancer who had received endocrine therapy previously [8]. Alpelisib recently received Food and Drug Administration (FDA) approval for this patient population [9]. Patients receiving alpelisib had an improved progression-free survival compared with fulvestrant alone by about 5 months and were at high risk of hyperglycemia-related adverse events. This modest benefit accompanied with significant tolerability challenges underscores the urgent need to identify potent, safe and tolerable PI3K inhibitors for breast cancer patients.

Elevated levels of reactive oxygen species (ROS) deregulated redox signaling and change in redox balance are frequent hallmarks of cancer progression [10]. ROS are highly reactive metabolites of oxygen including superoxide, hydrogen peroxide, hydroxyl radical, and hypochlorous acid. Cancer cells show persistent metabolic oxidative stress compared to normal cells, related to mitochondrial dysfunction. In addition, estrogens and metabolites of estrogen produce ROS in cancer cells [11]. We have developed a novel technology, called ROS-induced drug release (RIDR) that prevents inhibition of PI3K by a parent molecule until a high ROS environment is induced. The high ROS level leads to a chemical reaction that releases the known PI3K inhibitor to improve specificity of cancer cell activity [12,13]. We have focused on a small molecule, PI-103 [14,15,16] that produces tumor reduction in models of breast [17], lung cancer [18,19], and skin cancer. We observed increased selectivity of RIDR-PI-103 in the human acute myeloid leukemia cell line Kasumi-1 over normal cells, but no bioavailability data was performed in our previous study [20].

Anthracyclines, including doxorubicin, are a class of chemotherapeutics which have been FDA approved since the 1970s. Doxorubicin is a highly effective, first line chemotherapeutic agent used in the management of hematological and solid tumors and is cost-effective. Doxorubicin exerts its cytotoxic effect by intercalating DNA. Doxorubicin binds to DNA and topoisomerase 2 isoenzymes forming a ternary Top2-doxorubicin-DNA complex, which causes double-stranded DNA breaks [21]. Doxorubicin induces oxidative stress in cardiomyocytes [22,23] and can cause cardiomyopathy [24].

Our study indicates that low concentration of doxorubicin in combination with RIDR-PI-103 suppresses cancer cell growth and proliferation via inhibition of AKT signaling and induction of DNA damage response in breast cancer models.

## 2. Results

### 2.1. RIDR-PI-103 Has Increased Specificity in Breast Cancer Cells over Fibroblast and MCF10A Cells

We evaluated the toxicity effect of ROS-induced drug release 3-(4-morpholinopyrido[3′,2′:4,5]furo[3,2-d]pyrimidin-2-yl)phenol (RIDR-PI-103) (described in [20]) on normal fibroblasts, normal mammary epithelial breast cells MCF10A and estrogen receptor positive (ER+) MDA-MB-361, triple negative breast cancer (TNBC) MDA-MB-231 and HER2+ MDA-MB-453 breast cancer cell lines [25] using a 72 h 4,5-dimethylthiazol-2-yl)-2,5-diphenyltetrazolium bromide (MTT) cell proliferation assay. MDA-MB-453 cells contain the PIK3CA H1047R hot spot mutation and MDA-MB-361 contains PIK3CA E545K hot spot mutation [25]. We utilized MDA-MB-231 cells as these triple negative breast cancer (TNBC) cells have increased ROS compared to other breast cancer cells [26] and thus could have more activation of the ROS-activated scaffold drug. We found variable sensitivity of cells to RIDR-PI-103 (Figure 1). At concentrations of 50 µM (log 1.7 µM) or greater the three breast cancer cell lines showed growth inhibition to RIDR-PI-103, whereas normal fibroblasts or MCF10A cells did not respond appreciably to treatment (Figure 1). Dose response of PI-103 (0–5 µM) in MCF10A, normal fibroblasts and breast cancer cells is provided in Appendix A. Comparison of IC_50_ values of PI-103 and RIDR-PI-103 is tabulated in Appendix A.

### 2.2. Doxorubicin Induces ROS and Combination of RIDR-PI-103 and Doxorubicin Inhibits Breast Cancer Cell Viability

ROS levels were measured in MCF10A, MDA-MB-231, MDA-MB-361 and MDA-MB-453 cells with or without 30 min of doxorubicin (125 nM) treatment using 2′,7′ dichlorofluorescin diacetate (DCFDA) assay. The data indicated that MDA-MB-231 cells have the highest endogenous ROS versus MDA-MB-361, MDA-MB-453 and MCF10A cells. MCF10A cells have the lowest ROS levels compared to the three cancer cell lines. Doxorubicin induced ROS in all treatment groups (Figure 2A). In addition, we evaluated endogenous levels of antioxidants such as catalase, SOD1, SOD2, PRDX1 and GSR in fibroblast, MDA-MB-231, MDA-MB-361 and MDA-MB-453 cells. The data indicated that catalase expression was lower in all three breast cancer cell lines versus fibroblast. SOD1 levels are elevated in all three breast cancer cell lines versus fibroblast. There was variation in expression of SOD2, PRDX1 and GSR that didn’t correlate across the cancer cell lines and fibroblast (Appendix A). We next evaluated the combination of RIDR-PI-103 and doxorubicin in MDA-MB-231, MDA-MB-361 and MDA-MB-453 cell lines. We used serial concentrations of doxorubicin in combination with RIDR-PI-103 (10–30 µM) based on the IC_50_ values of doxorubicin in different cancer cell lines. We found that 10, 15 and 30 µM RIDR-PI-103 enhances the cytotoxicity of the anthracycline doxorubicin in MDA-MB-231, MDA-MB-361 and MDA-MB-453 (Figure 2B–D). We further assessed the effect of the ROS scavenger, N-Acetyl Cysteine (NAC) alone or with RIDR-PI-103 +/- doxorubicin in MDA-MB-231 and MDA-MB-453 cells. We observed that NAC rescued the antiproliferative effects of RIDR-PI-103 and doxorubicin in MDA-MB-231 cells (Appendix A) but not in MDA-MB-453 cells (Appendix A). Our data indicate that MDA-MB-231 cells contain more endogenous ROS than MDA-MB-453 (Figure 2A). We speculate that with increased ROS, MDA-MB-231 cells may be more sensitive to treatment with the ROS scavenger NAC, allowing for rescue of treatment with RIDR-PI-103 and doxorubicin. We also analyzed the effect of docetaxel [27], part of the taxane class that act as anti-microtubule agents, in combination with RIDR-PI-103. The data indicated that docetaxel was less effective in combination with RIDR-PI-103 to suppress breast cancer cell growth (Appendix A). IC50 values of DOXO, DOXO ± 10 µM RIDR, DOXO ± 15 µM RIDR and DOXO ± 30 µM RIDR in MDA-MB-231, MDA-MB-361 and MDA-MB-453 cell lines are provided in Appendix A. Notably, in MDA-MB-231 cells the IC_50_ value was reduced by more than one half comparing doxorubicin and RIDR-PI-103 combination versus doxorubicin alone (Appendix A).

### 2.3. Doxorubicin in Combination with RIDR-PI-103 Suppresses Breast Cancer Cell Proliferation

We speculated that the combination of doxorubicin and RIDR-PI-103 could effectively inhibit breast cancer cell proliferation. Accordingly, MDA-MB-231, MDA-MB-361 and MDA-MB-453 cells were treated with doxorubicin (125 nM) and RIDR-PI-103 (10 µM) every alternative day and stained after 10–12 days using a two-dimensional crystal violet assay. The data indicated that the combination of doxorubicin and RIDR-PI-103 significantly reduced cancer cell proliferation versus DMSO or single agent doxorubicin or RIDR-PI-103 in all three cell lines (Figure 3A–C).

### 2.4. Doxorubicin in Combination with RIDR-PI-103 Suppresses Matrigel Colony Formation

Examination of cells grown on a basement membrane of matrigel indicated that combination doxorubicin (125 nM) and RIDR-PI-103 (10 µM) inhibit colony formation better than either single agent. Our data showed that the combination of doxorubicin with RIDR-PI-103 significantly suppressed colony formation on matrigel compared to individual drugs or DMSO control in MDA-MB-231, MDA-MB-361 and MDA-MB-453 cells. MDA-MB-453 cells were most sensitive to the combination of doxorubicin and RIDR-PI-103 marked by significantly reduced colonies versus single agent RIDR-PI-103 or doxorubicin or DMSO. MDA-MB-231 cells displayed aggressive growth with vehicle treatment and had a statistically significant reduction in the colony formation with combination treatment versus either single agent treatment (Figure 4A–C).

### 2.5. Doxorubicin and RIDR-PI-103 Suppress Breast Cancer Cell Migration

Transwell migration assay was performed to assess the effect of doxorubicin and RIDR-PI-103 on breast cancer cell migration. The data indicated that the combination of doxorubicin and RIDR-PI-103 inhibit breast cancer cell migration better compared to single agent doxorubicin, RIDR-PI-103 or DMSO control. Our cell migration data also showed that in all three breast cancer cell lines co-treatment of doxorubicin and RIDR-PI-103 resulted in statistically significant reduction in migration compared to either single agent (Figure 5A–C).

### 2.6. The Combination of Doxorubicin and RIDR-PI-103 Results in Enhanced Inhibition of PI3K Signaling and Activates DNA Damage Response

We sought to examine RIDR-PI-103 activity via a series of western blots as shown in Figure 6. RIDR-PI-103 targets PI3K and the downstream AKT signaling pathway. There was no effect of either doxorubicin or RIDR-PI-103 in phosphorylation of AKT at Thr308 (data not shown). In MDA-MB-231 breast cancer cells, a combination of doxorubicin with RIDR-PI-103 suppressed AKTSer473 phosphorylation as compared to single agent doxorubicin or RIDR-PI-103 (Figure 6). Doxorubicin leads to DNA damage. The protein p53 is an imperative tumor suppressor molecule which plays an important role in DNA damage signaling and various genomic alterations [28]. Activation of p53 leads to cell cycle arrest or apoptosis. DNA damage induces phosphorylation of p53 at Ser15 and Ser20 and leads to a reduced interaction between p53 and its negative regulator, the oncoprotein MDM2 [29,30]. Our data indicated that doxorubicin (125 nM) in combination with RIDR-PI-103 (10 µM) induced p53 phosphorylation at Ser15 in MDA-MB-231 breast cancer cell lines with similar results in MDA-MB-361 and MDA-MB-453 cells. However, we didn’t observe p53 post-translational modification activation in vehicle (DMSO) or doxorubicin treatment alone. We further examined the effect of the combination on cell cycle arrest. CHK1 kinase is downstream of ATM/ATR kinase pathway and plays a crucial role in DNA damage checkpoint control and tumor suppression. Doxorubicin in combination with RIDR-PI-103 induced p-CHK1 compared to either single agent. Checkpoint kinase 2 protein, (CHK2) is also downstream of ATM/ATR. CHK2 contain a series of seven serine/threonine residues (Ser19, Thr26, Ser28, Ser33, Ser35, Ser50, and Thr68) each followed by glutamine (SQ or TQ motif) [31]. Examining the phosphorylation of Thr68 of CHK2, our results indicate that doxorubicin, RIDR-PI-103, and the combination increased phosphorylation of CHK2 at T68 most notably in MDA-MB-361 and MDA-MB-453 versus vehicle control. All original images from western blots are included in Appendix A.

### 2.7. Pharmacokinetics Profile of RIDR-PI-103

In this initial PK study in mice, RIDR-PI-103 (20 mg/kg) was administered intraperitoneally and blood samples were collected over a period of 96 h. The concentration of 20 mg/kg was chosen based on the ability to solubilize RIDR-PI-103 in 40% propylene glycol with 60% injectable saline. The mean plasma concentration-time profile was then analyzed using Phoenix^®^ WinNonlin^®^ v8.2 using both a compartmental and non-compartmental analysis (NCA). As shown (Figure 7A,B), the plasma concentration-time profile fitted a one-compartment model. The key PK parameters are listed in Table 1. As shown, both methods of PK analysis yielded similar results. RIDR-PI-103 has a maximal plasma concentration (C_max_) of 201.5 ng/mL (0.43 µM), which was achieved at the time taken to reach the maximum concentration (T_max_) of 1.44 h. The elimination half-life of RIDR-PI-103 was 9.4 h (Table 2), aligning with the blood sample schedule employed (up to 96 h, approximately a period equivalent to 10 half-lives) facilitating complete characterization of the elimination profile. Based on these studies, RIDR-PI-103 has a large volume of distribution (Vd), 89 L/kg.

## 3. Discussion

The PI3K/AKT/mTOR pathway is an important therapeutic target for treatment of breast cancer [2,32]. Several PI3K inhibitors are approved by the FDA for different cancers: Alpelisib in HR+ and HER2/neu negative breast cancer [11], idelalisib for leukemia [33] and two types of lymphoma [34,35], duvelisib for chronic lymphocytic leukemia/small lymphocytic lymphoma [36]. Several other PI3K inhibitors are also in different stages of clinical development [37,38,39,40]. Drugs targeting PI3K or mTOR catalytic activity are toxic, due to the physiological roles of PI3K/mTOR signaling in basic cellular processes in tissues throughout the body, including protein translation, intracellular trafficking, autophagy, and metabolism. Toxicity that occurs in patients includes hyperglycemia as genes encoding most glycolytic enzymes are under transcriptional control by PI3K/AKT [7].We designed a novel PI-103 prodrug (RIDR-PI-103) such that the PI3K inhibitor PI-103 would only be released under high oxidative stress conditions found in the tumor milieu. Thus, RIDR-PI-103 is designed to only inhibit PI3K in the tumor microenvironment under high oxidative stress conditions. This would circumvent the toxicity observed with global inhibition of PI3K throughout the body using current PI3K inhibitors. The study herein describes the efficacy of RIDR-PI-103 in breast cancer cells along with initial PK profile in a mouse model.

Enhanced ROS in non-transformed cells or breast cancer cells could have pro-tumorigenic effects via damaging nucleic acids and inducing genomic instability. Breast cancer subtypes demonstrate differential ROS production and susceptibility to antioxidant treatment. TNBC cells have increased ROS levels compared to non-tumorigenic or ER+/luminal breast cancer cells. TNBCs have higher oxidation states marked with enhanced ROS marker (dihydroethidium and MitoSox) staining compared to ER+ and non-tumorigenic control [26]. This is consistent with our findings demonstrating that TNBC MDA-MB-231 cells have high ROS compared to non-tumorigenic MCF10A cells. Doxorubicin produces ROS in vivo. Further, ROS plays a mechanistic role in the cardiotoxicity induced by doxorubicin involved in breast cancer treatment [41,42]. We found that doxorubicin at a concentration 125 nM induced ROS in breast cancer cell lines (MDA-MB-231, MDA-MB-361 and MDA-MB-453) which was higher than that induced in MCF10A cells (Figure 2A).

The PI3K pathway is significant in maintaining genomic stability by involving DNA replication and cell cycle regulation [43]. For example mTOR inhibition, downstream of PI3K, has been shown to activate p-CHK2 T68, an indicator of ATM-CHK2 checkpoint activation [44]. We examined the effect of combination doxorubicin and RIDR-PI-103 in Akt and DNA damage response signaling. Our data showed that the combination of doxorubicin and RIDR-PI-103 activated p-53 in three cancer cell lines. We observed that doxorubicin and RIDR-PI-103 combined upregulated p-CHK1S345 and p-CHK2T68 in MDA-MB-361 and MDA-MB-453 cells. PI3K inhibition can induce DNA damage via nucleoside depletion notably in cells with genetic aberrations in p53 and BRCA1 [45]. MDA-MB-453 cells have been reported to have a mutation in exon 11 of p53. MDA-MB-231 cells contains a R280K p53 mutation and MDA-MB-361 cells contains a p53 mutation in exon 4 [46]. Thus, inhibition of PI3K with RIDR-PI-103 in these breast cancers could reduce nucleotide triphosphates, resulting in DNA damage and activation of DNA damage response. The exact mechanism(s) by which RIDR-PI-103 can activate a DNA damage response as indicated by increased p-p53, p-CHK1 and p-CHK2 remain to be explored in future studies.

Recently, a bioisostere of PI-103 has been developed with a structural modification containing a boronate in place of a phenolic hydroxyl group with the goal to enhance bioavailability [47]. The boron-containing PI-103 bioisostere demonstrated improved bioavailability relative to PI-103. RIDR-PI-103 differs from the boron-containing compound in that RIDR-PI-103 is designed to only release PI-103 under controlled oxidative stress conditions when ROS levels are high, present in the tumor microenvironment milieu. Our goal with the design of RIDR-PI-103 is to circumvent toxicity observed with systemic inhibition of PI3K signaling throughout the body [48].

Herein, we performed an initial PK study meant to estimate some of the drug disposition parameters such as the peak plasma levels, elimination half-life, apparent volume of distributions and total clearance. Since we have not performed a PK study following an intravenous (i.v.) administration, the bioavailability following the intraperitoneal (i.p.) administration cannot be ascertained at this point. The peak plasma concentration at a dose of 20 mg/kg was 0.43 µM. The apparent volume of distribution (89 L) is suggestive of extensive drug distribution while the half-life of approximately 10 h suggesting that the systemic drug elimination is relatively slow. Thus, these derived PK data imply that an efficacious dosing regimen can be developed to yield effective tissue-specific drug levels.

Our study has limitations. The scope of the study focuses on in vitro studies with one in vivo pharmacokinetic study. The PK study was performed in tumor-free mice. In follow up studies, we will first derive a correlation between extracellular (cell culture media) and intracellular concentrations of RIDR-PI-103 which will then reflect on tumor-tissue specific concentrations needed for efficacy in animal models. The PK data derived here can then be employed for designing effective dosing regimen (dose and dosing frequency) by simulating PK profiles assuming linear PK (i.e., no change in the half-life, clearance and volume of distribution). Additional PK studies will measure the amount of PI-103 present, as it will be important to assess the conversion of pro-drug to biologically active drug. Future PK studies will employ a higher dose of RIDR-PI-103 with the initial dose described here using only 20 mg/kg RIDR-PI-103.

Future studies will examine the in vivo efficacy of single-agent RIDR-PI-103 compared to parent drug PI-103 in breast tumor models. There is evidence that inhibition of PI3Kɣ protects from anthracycline toxicity [49]. Additional in vivo efficacy studies will examine if RIDR-PI-103 has a cardioprotective effect in combination with doxorubicin as PI-103 is a pan-PI3K inhibitor with an IC50 against PI3Kɣ of 15 nM [50]. Overall, these results are promising and further help in PK modeling and deduction of a safe and efficacious starting dose for efficacy studies utilizing novel RIDR-PI-103 in breast cancer model.

## 4. Materials and Methods

### 4.1. Cell Culture and Inhibitors

Human breast cancer cells (MDA-MB-231, MDA-MB-361 and MDA-MB-453) were obtained from American Type Culture Collection (ATCC). MCF10A cells were a generous gift from Dr. Nalinikanth Kotagiri, College of Pharmacy, University of Cincinnati. NH-fibroblast was obtained from Dr. Edward J. Merino, Department of Chemistry, University of Cincinnati. MDA-MB-231 cells were maintained in RPMI media supplemented with 10% Fetal Bovine Serum (FBS). MDA-MB-361, MCF10A and MDA-MB-453 cells were cultured in DMEM media with 10% FBS. NH-fibroblasts cells were maintained in MDME media with 10% FBS and supplemented with growth factors such as insulin, hydrocortisone and epidermal growth factor (EGF). RIDR-PI-103 was synthesized in collaboration with Dr. Edward J. Merino [20]. Doxorubicin (Cat# D-4000) and docetaxel (Cat# D-1000) were obtained from LC Laboratories (Woburn, MA, USA). All these compounds were prepared in DMSO solvent.

### 4.2. Immunoblot Analysis

Cells were lysed in RIPA buffer (Thermo Fisher Scientific, Waltham, MA, USA, Cat#BP-115) supplemented with protease and phosphatase inhibitors (Thermo Fisher Scientific, Cat#88669). Immunoblotting was performed using the following antibodies: p-AktS473 (Cell Signaling Technology, Danvers, MA, USA Cat#4060), Akt (Cell Signaling Technology, Cat#9272), p-P53S15 (Cell Signaling Technology, Cat#9286), P53 (Cell Signaling Technology, Cat#2527), p-CHK1S345 (Cell Signaling Technology, Cat#2348), CHK1 (Cell Signaling Technology, Cat#2360), p-CHK2T68 (Cell Signaling Technology, Cat#2661), CHK2 (Cell Signaling Technology, Cat#6334), and Actin (Santa Cruz Biotechnology, Dallas, TX, USA, Cat# SC-1616). Peroxidase-conjugated secondary antibodies (Santa Cruz Biotechnology) were used and protein signals were detected using Pierce ECL western blotting substrate (Thermo Fisher Scientific, Cat# 32106).

### 4.3. Cell Viability Assay

Growth kinetics of fibroblasts, MCF10A, MDA-MB-231, MDA-MB-361 and MDA-MB-453 cells was determined by the 4,5-dimethylthiazol-2-yl)-2,5-diphenyltetrazolium bromide (MTT) assay. Briefly, 2 × 10^4^ cells/well were seeded in 96 well plates in triplicate. After 72 h of treatment with RIDR-PI-103 (0–110 µM) and PI-103 (0–5 µM), media was substituted with 50 mg/mL MTT solution (Sigma Aldrich, St. Louis, MO, USA) and absorbance was recorded at 570 nm using SPECTRAmax PLUS Microplate Spectrophotometer Plate Reader (Molecular Devices Corporation, San Jose, CA, USA) and expressed as the mean of triplicates relative to vehicle (DMSO) control together with standard error of mean (SEM). In separate experiments, MDA-MB-231, MDA-MB-361 and MDA-MB-453 cells were treated with a series of concentrations using doxorubicin in the presence or absence of RIDR-PI-103 (10–30 µM). MDA-MB-231 cells were treated with 500–5000 nM, MDA-MB-361 with 50–450 nM and MDA-MB-453 with 100–4000 nM of doxorubicin for 72 h. MDA-MB-231, MDA-MB-361 and MDA-MB-453 cells were treated with a series of concentrations using docetaxel in the presence or absence of RIDR-PI-103 (10, 15 or 30 µM). MDA-MB-231 cells were treated with 50–50,000 pM docetaxel. MDA-MB-361 cells were treated with 10–90 nM of docetaxel. MDA-MB-453 cells were subjected 50–10,000 pM of docetaxel. The cell proliferation was analyzed after 72 h and represented as mean of triplicate values relative to DMSO control. The bar graph generated using graph pad prism 7 (GraphPad Software, Inc., La Jolla, CA, USA).

### 4.4. Cell Proliferation Assay

MDA-MB-231, MDA-MB-361 and MDA-MB-453 cells were seeded at a density of 5 × 10^4^ cells/well in 6 well plates (Thermo Fisher Scientific) in triplicate. Complete media containing 125 nM doxorubicin or 10 µM RIDR-PI-103 alone or combination of both respectively was replaced every alternate day and cells were stained with 0.5% crystal violet in methanol within 7–10 days. The intensities were measured using an Odyssey infrared system (LI-COR, Lincoln, NE, USA). The values are expressed as mean of intensities obtained from 3 independent experiments.

### 4.5. Cell Migration Assay

MDA-MB-231, MDA-MB-361 and MDA-MB-453 cells were seeded at a density of 5 × 10^4^ cells/well in 6 well plates and treated with vehicle (DMSO), 125 nM doxorubicin and 10 µM RIDR-PI-103 or combination of both for 8 h. After 8 h, cells were counted and 2 × 10^4^ cells/well were added to the upper chamber of transwell and incubated for 24–48 h. After 24–48 h, migrated cells in the lower chamber were stained with 0.5% crystal violet and images were captured from three areas under phase contrast microscope. The intensities from three areas collected from three independent experiments were measured using ImageJ and expressed as % of control and represented graphically.

### 4.6. Measurement of Intracellular ROS

MCF10A, MDA-MB-231, MDA-MB-361 and MDA-MB-453 cells (2 × 10^4^) were plated and treated with or without 125 nM of doxorubicin for 30 min at 37 °C. The cells were resuspended in dichlorodihydrofluorescein diacetate (DCFH-DA) for 30 min at analyzed using the CellQuest software v4.0.1 (Becton Dickinson and Company, Franklin Lakes, NJ, USA) to measure the ROS levels. The DCFH-DA levels were measured using a SPECTRAmax PLUS Microplate Spectrophotometer Plate Reader (Molecular Devices Corporation).

### 4.7. Matrigel Colony Formation

Three-dimensional (3D) growth assays were performed in growth factor-reduced matrigel (BD Biosciences, San Jose, CA, USA) where 96 well plates were coated with 80 µL of matrigel/well. MDA-MB-231, MDA-MB-361 and MDA-MB-453 cells (1 × 10^4^/well) were plated and incubated at 37 °C for 24 h. Cells were treated with vehicle (DMSO), 10 µM RIDR-PI-103 or 125 nM doxorubicin or combination of both every alternate day. After 10 days of incubation, colonies were visualized and photographs were captured from 3 random fields under microscope (Nikon, Road Melville, NY, USA) at 10× magnification. The areas of the colonies were measured by ImageJ and represented as mean areas normalized to DMSO control. The experiments were repeated 3 times to confirm the results.

### 4.8. Analytical Methods

A high performance liquid chromatography (HPLC) was used to analyze serial dilutions of RIDR-PI-103 (1000, 500, 250, 125, 12.5 ng/mL) as described previously [28]. Briefly, we employed a Waters column (Milford, MA, USA) and the samples were eluted using mobile phases A: 95% water + 5% acetonitrile and B: 5% water + 95% acetonitrile. The HPLC was performed using a gradient method, with 0% B for 4 min and 95% B for over 15 min. The flow was 1 mL/min. A detection wavelength of 250 nm was used on Waters 2487 Dual Wavelength Absorbance Detector. Once the HPLC method was developed, we then set up a liquid-liquid extraction method for RIDR-PI-103 from mouse plasma. Extractions were performed using standard liquid-liquid extraction process with methyl tert-butyl ether (MTBE) used as extraction solvent. The samples were reconstituted in 100 µL of PBS: DMSO (99:1 *v*/*v*) and the pH was adjusted to 2 using 1 M hydrochloric acid (HCl). The extraction efficiency was observed to be ~40% with the limit of detection between 1.25 ng/mL and 1250 ng/mL.

### 4.9. Pharmacokinetic Analysis

The plasma concentration-time profile of RIDR-PI-103 was analyzed using Phoenix^®^ WinNonlin^®^ v8.2 (Certara L.P. (Pharsight), St. Louis, MO, USA). Regression analysis of the data suggested that a compartment model best fitted the data. This was based on observed goodness of fit (random distribution of the residual around the predicted curve) and the Akaike Information Criterion (AIC), and the Schwarz Criterion (SC). The important PK parameters derived using this approach included elimination half-life (T_½_), time required to achieve peak plasma concentrations (T_max_), total Area Under the concentration-time Curve (AUC0–∞) and systemic oral clearance (CL/F) and apparent volume of distribution (Vd/F). “F” is the bioavailability following the intraperitoneal (i.p) administration and cannot be estimated in the absence of intravenous (i.v.) administration. We also employed the non-compartmental analysis (NCA) to verify the PK parameters derived using the one-compartment model. The NCA utilizes the trapezoidal rule for AUC (0-tlast) determination over the blood sample collection period. The regression analysis of the last four concentration-time values (24, 48, 72 and 96 h) was done to determine the elimination rate constant (Kel). PK parameters such as AUC (0–∞), Cl/F, Vd/F and the elimination half-life were then computed as shown below:AUC (0–∞) = AUC (0-tlast) + Clast/kel(1)
Cl/F = Dose/AUC (0–∞)(2)
Vd/F = (Cl/F)/kel(3)

### 4.10. Animal Studies

Female C57BL/6J mice at 4 weeks were used with *n* = 3 per time point. Time points for blood collection were based on previous findings regarding the in vitro microsomal metabolic stability of RIDR-PI-103 [20]. RIDR-PI-103 was formulated using a mixture of 40% propylene glycol with 60% injectable saline in which RIDR-PI-103 was soluble in solution and not a suspension. The stability of RIDR-PI-103 in this formulation was ascertained by measuring drug content over 7 days. RIDR-PI-103 (dose = 20 mg/kg) was injected intraperitoneally in all mice at the start of the experiment. Blood collection was done via cardiac puncture under anesthesia at 0, 0.5, 4, 6, 24, 48, 72, 96 h post injection. Plasma was isolated from the blood samples by centrifugation and was stored at −80 °C until further analysis. Institutional Animal Care and Use Committee (IACUC), University of Cincinnati ethically approved the in vivo mouse experiment under protocol AM02-19-08-28-01, approved on 27 July 2020.

### 4.11. qPCR Analysis of Antioxidant mRNAs

Total RNA was extracted from fibroblasts, MDA-MB-231, MDA-MB-361 and MDA-MB-453 cells using the RNA extraction mini kit (Qiagen, Germantown, MD, USA) according to the manufacturer’s instruction. Two-step RT-qPCR was performed to assess the mRNA level of SOD1, SOD2, CAT, PRDX1 and GSR. First strand cDNA was synthesized using iScriptTM cDNA Synthesis Kit (BioRad, Hercules, CA, USA). q-PCR was set up using CFX96 Real-Time System (BioRad). Actin was used as an internal control. Paired primer sequences used were SOD1: 5′-GGTGGGCCAAGGATGAAGAG-3′ (forward) and 5′-CCACAAGCCAACGACTTCC-3′ (reverse); SOD2: 5′-GCTCCGGTTTTGGGGTATCTG-3′ (forward) and 5′-GCGTTGATGTGAGGTTCCAG-3′ (reverse); CAT 5′-TGGAGCTGGTAACCCAGTAGG-3′ (forward) and 5′-CCTTTGCCTTGGAGTATTTGGTA-3′ (reverse); PRDX1 5′- CCACGGAGATCATTGCTTTCA-3′ (forward) and 5′-AGGTGTATTGACCCATGCTAGAT -3′ (reverse); GSR: 5′-CACTTGCGTGAATGTTGGATG-3′ (forward) and 5′-TGGGATCACTCGTGAAGGCT-3′ (reverse); and Actin: 5′AAGGAGCCCCACGAGAAAAAT-3′ (forward); 5′- ACCGAACTTGCATTGATTCCAG-3′.

### 4.12. Statistical Analysis

Data are shown as the mean ± standard error of mean (SEM) and representative of at least three independent experiments unless otherwise indicated. Statistical analysis among groups using the two-tailed Student’s *t*-test, one-way analysis of variance, *p* < 0.05 was considered statistically significant.

## 5. Conclusions

We describe a ROS-activated prodrug (RIDR-PI-103) that in combination with doxorubicin inhibits breast tumor growth and migration. Our data indicated that doxorubicin with RIDR-PI-103 downregulated Akt signaling and activated DNA damage response signals.

## Figures and Tables

**Figure 1 ijms-22-02088-f001:**
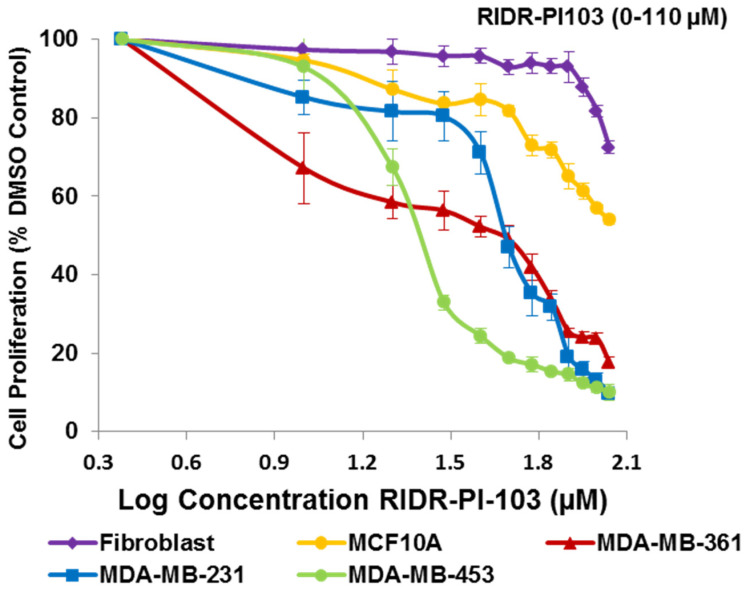
Dose response of RIDR-PI-103 in MCF10A, normal fibroblasts and breast cancer cells. NH fibroblasts, MCF10A, MDA-MB-231, MDA-MB-361 and MDA-MB-453 cells were plated in triplicate in 96 well plates and treated with RIDR-PI-103 (0–110 µM) for 72 h. The data was analyzed and presented using GraphPad Prism v7 (*n* = 3 independent experiments performed in triplicate ± SEM). IC50 values determined using graph pad prism as: Fibroblasts = N.D.; MCF10A = 78.6 µM; MDA-MB-231 = 47.3 µM, MDA-MB-361 = 43.01 µM and MDA-MB-453 = 49.05 µM.

**Figure 2 ijms-22-02088-f002:**
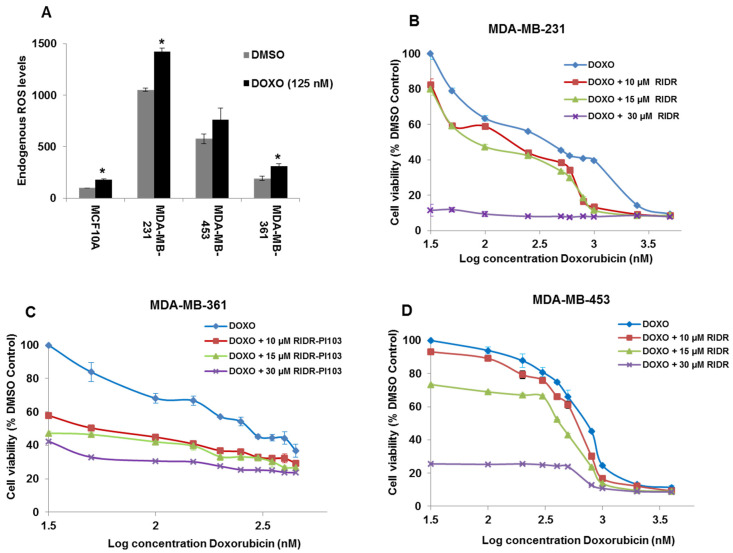
Doxorubicin induces ROS and combination of RIDR-PI-103 and Doxorubicin inhibit breast cancer cell viability. (**A**) Reactive oxygen species (ROS) levels were measured by DCFDA assay where MCF10A, MDA-MB-231, MDA-MB-361 and MDA-MB-453 cells were treated with or without 125 nM doxorubicin (DOXO) for 30 min. Dimethyl sulfoxide (DMSO) was used as vehicle control. Results are represented as mean ± SEM from 3 biological triplicates and represented graphically. * indicates *p* < 0.05 compared to DMSO treatment. (**B**) MDA-MB-231 cells (2 × 10^4^ cells/well) were seeded in 96 well plates in triplicate and treated with a serial concentration of doxorubicin (DOXO) in presence of 10 µM, 15 µM and 30 µM RIDR-PI-103 (RIDR) for 72 h. The concentration of doxorubicin used was 500–5000 nM for MDA-MB-231. (**C**) MDA-MB-361 cells (2 × 10^4^ cells/well) were seeded in 96 well plates in triplicate and treated with a serial concentration of doxorubicin (DOXO) in presence of 10 µM, 15 µM and 30 µM RIDR-PI-103 (RIDR) for 72 h. The concentration of doxorubicin used was 50–450 nM for MDA-MB-361. (**D**) MDA-MB-453 cells (2 × 10^4^ cells/well) were seeded in 96 well plates in triplicate and treated with a serial concentration of doxorubicin (DOXO) in presence of 10 µM, 15 µM and 30 µM RIDR-PI-103 (RIDR) for 72 h. The concentration of doxorubicin used was 100–4000 nM for MDA-MB-453. All the cells were treated with MTT (5 mg/mL) for 4 h and absorbance read at 570 nm in a microtiter plate reader (*n* = 3 independent experiments performed in triplicate ± SEM).

**Figure 3 ijms-22-02088-f003:**
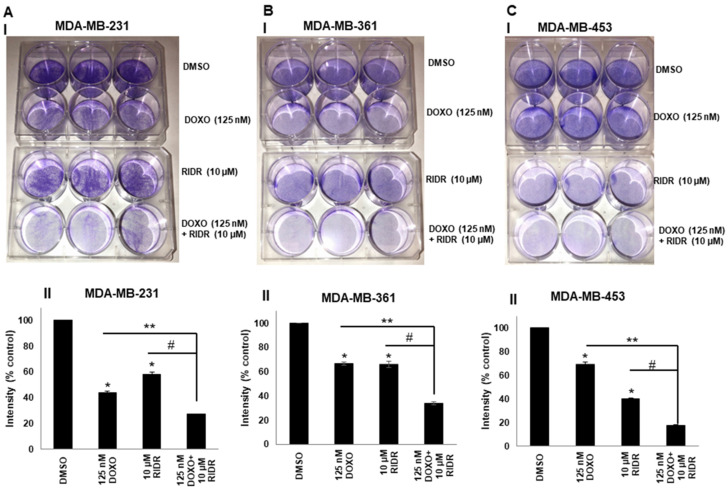
Combination of doxorubicin and RIDR-PI-103 inhibit breast cancer cell proliferation. (**A**) MDA-MB-231 cells (5 × 10^4^ cells/well) were plated and treated with media containing vehicle (DMSO), doxorubicin (125 nM) or RIDR-PI-103 (10 µM) and indicated combinations every second day and stained with crystal violet within 7–10 days (panel I). The intensities are represented as mean; Error bars: SEM (*n* = 3 independent experiments performed in triplicate). * *p* < 0.05 versus DMSO and **,^#^
*p* < 0.05 versus individual treatment as indicated (panel II). (**B**) MDA-MB-361 cells (5 × 10^4^ cells/well) were plated and treated with media containing vehicle (DMSO), doxorubicin (125 nM) or RIDR-PI-103 (10 µM) and indicated combinations every second day and stained with crystal violet within 7–10 days (panel I). The intensities are represented as mean; Error bars: SEM (*n* = 3 independent experiments performed in triplicate). * *p* < 0.05 versus DMSO and **,^#^
*p* < 0.05 versus individual treatment as indicated (panel II). (**C**) MDA-MB-453 cells (5 × 10^4^ cells/well) were plated and treated with media containing vehicle (DMSO), doxorubicin (125 nM) or RIDR-PI-103 (10 µM) and indicated combinations every second day and stained with crystal violet within 7–10 days (panel I). The intensities are represented as mean; Error bars: SEM (*n* = 3 independent experiments performed in triplicate). * *p* < 0.05 vs. DMSO and **,^#^
*p* < 0.05 versus individual treatment as indicated (panel II).

**Figure 4 ijms-22-02088-f004:**
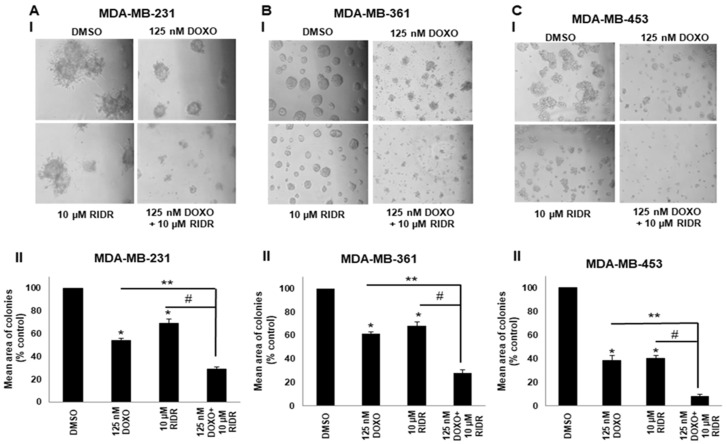
Doxorubicin and RIDR-PI-103 inhibit colony formation on matrigel. (**A**) MDA-MB-231 cells (2 × 10^4^ cells/well) were seeded on a basement membrane of matrigel and treated with vehicle (DMSO), doxorubicin (125 nM), RIDR-PI-103 (10 µM) or combination every second day. (**B**) MDA-MB-361 cells (2 × 10^4^ cells/well) were seeded on a basement membrane of matrigel and treated with vehicle (DMSO), doxorubicin (125 nM), RIDR-PI-103 (10 µM) or combination every second day. (**C**) Equal number of MDA-MB-453 cells were seeded on a basement membrane of matrigel and treated with vehicle (DMSO), doxorubicin (125 nM), RIDR-PI-103 (10 µM) or combination every second day. Phase contrast images of acini for all cell lines were captured at 10× magnification (panel I) and the average size of each cellular structure was quantified and expressed as mean of areas ± SEM, (*n* = 3 independent experiments performed in triplicate). * *p* < 0.05 vs. DMSO and **,^#^
*p* < 0.01 versus respective treatment groups (panel II).

**Figure 5 ijms-22-02088-f005:**
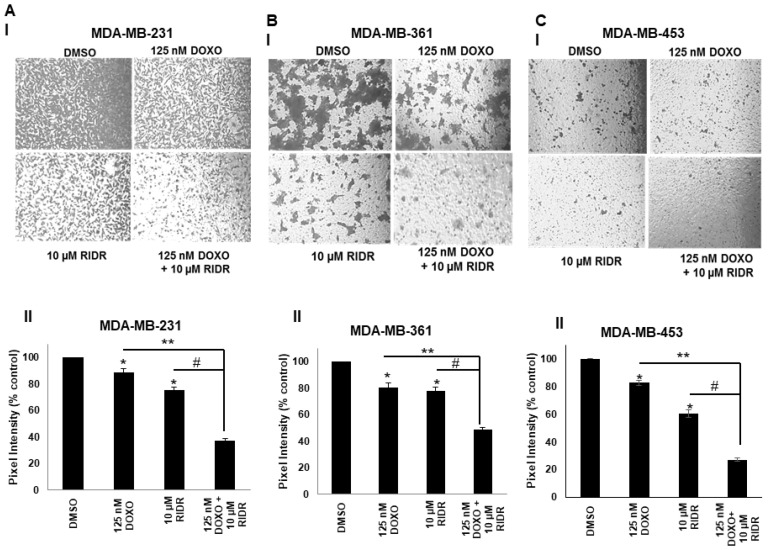
Doxorubicin and RIDR-PI-103 suppress breast cancer cell migration. (**A**) MDA-MB-231 cells were seeded at a density of 5 × 10^4^ cells/well and treated with vehicle (DMSO), doxorubicin (125 nM), RIDR-PI-103 (10 µM) and indicated combination in serum free media for 24 h. (**B**) MDA-MB-361 cells were seeded at the same density and treated with indicated combination in serum free media for 24 h. (**C**) MDA-MB-453 cells were seeded at a density of 5 × 10^4^ cells/well and treated with vehicle (DMSO), doxorubicin (125 nM), RIDR-PI-103 (10 µM) and indicated combination in serum free media for 48 h. 10% FBS was added as a chemoattractant in the lower chamber. After 24–48 h, the migrated cells were stained and intensities were measured using ImageJ and expressed as % of control (panel I). The bar graphs are represented as mean; Error bars: SEM (*n* = 3 independent experiments performed in triplicate), *p*-value < 0.05 for comparison of * to DMSO, ** for comparison of doxorubicin vs. doxorubicin and RIDR-PI-103, ^#^ for comparison of RIDR-PI-103 vs. doxorubicin and RIDR-PI-103 (panel II).

**Figure 6 ijms-22-02088-f006:**
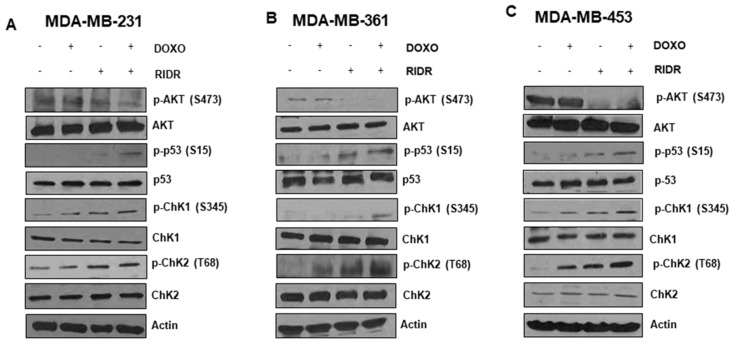
Doxorubicin in combination RIDR-PI-103 inhibits AKT and downstream signaling. Equal number of MDA-MB-231 (**A**), MDA-MB-361 (**B**) and MDA-MB-453 (**C**) cells were treated with vehicle (DMSO), RIDR-PI-103 (10 µM), doxorubicin (125 nM) and combination of both for 24 h and cell lysates were collected and immunoblotted using p-AKTSer473, AKT, p-P53Ser15, P53, p-CHK1S345, CHK1, p-CHK2T68 and CHK2 antibodies. Actin served as loading control.

**Figure 7 ijms-22-02088-f007:**
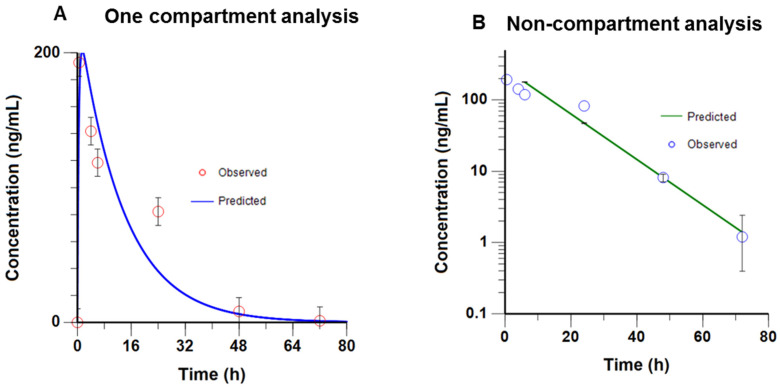
Pharmacokinetic profile of RIDR-PI-103 (**A**,**B**). A single dose of RIDR-PI-103 (20 mg/kg) was administered intraperitoneally in mice and serial blood samples were collected over a period of 96 h (*n* = 3 per time point). Concentration-time profile for RIDR-PI-103 in mice plasma using one-compartmental and non-compartmental analysis.

**Table 1 ijms-22-02088-t001:** The key PK parameters for the plasma concentration-time profile fitted a one-compartment model for RIDR-PI-103.

PK Parameters	Observed Values
V/F	89 L/kg
K01	2.5 h^−1^
K10	0.075 h^−1^
AUC (0–72 h)	2979.8 h·ng/mL
Elimination T_1/2_	9.18 h
Cl/F	6.7 L/h/kg
T_max_	1.44 h
C_max_	201.5 ng/mL

**Table 2 ijms-22-02088-t002:** The key PK parameters for the plasma concentration-time profile fitted a non-compartment model for RIDR-PI-103.

PK Parameters	Observed Values
V/F	89 L/kg
λz	0.073 h^−1^
K10	0.073 h^−1^
AUC (0–72 h)	3897.1 h·ng/mL
Elimination T_1/2_	9.44 h
Cl/F	5.1 L/h/kg
T_max_	0.5 h
C_max_	192.8 ng/mL

## Data Availability

Not applicable.

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
