# Peer review of "Phosphoinositide 3-Kinase (PI3K) Reactive Oxygen Species (ROS)-Activated Prodrug in Combination with Anthracycline Impairs PI3K Signaling, Increases DNA Damage Response and Reduces Breast Cancer Cell Growth"

_ijms, 2021, doi:10.3390/ijms22042088_

Round 1

Reviewer 1 Report

Review of ijms-1087975

The manuscript by Mishra et al describes the activity of a novel ROS-activated prodrug (RIDR-PI-103) of the PI3K inhibitor PI-103 in breast cancer cells both as monotherapy and in combination with doxorubicin. Whilst the paper is of potential interest, I found it lacking in several control experiments that made the data harder to evaluate, and felt that a greater evaluation of RIDR-PI-103 as a monotherapy (rather than in combination with doxorubicin) would have been more interesting.

Specific points:

(1) Figure 1:

                1.1 – Can the GI50 values (concentration that induces 50% growth inhibition) be calculated and tabulated.

                1.2 – how do these compare to the GI50 values for PI-103?

                1.3 – please use an x-axis scale with standard, easy to read values along the bottom. Log scale with 0.1, 1, 10, 100 etc is preferable.

                1.4 – have the authors determined the amount of RIDR-PI-103 converted to PI-103 in each cell line?

(2) From figure 2, given that MDA-MB-453 has less ROS than MB-231, why did the authors think that MB-453 are more sensitive to RIDR-PI-103 than MB-231s?

(3) Figure 2:

                3.1 – define what symbols on fig 2A refer to.

3.2 – does the increased ROS generated by dox treatment result in increased conversion of RIDR-PI-103 to PI-103?

                3.3 – the dose response curves shown here have, in my opinion, been incorrectly set up. Based on the authors suggestion that Dox increases ROS and therefore sensitises cells to RIDR-PI-103, I would have expected a concentration response of RIDR-PI-103 with fixed concentrations of Dox (not the other way round as shown).

                3.4 – GI50 values should be calculated and presented. From the data shown, the effects appear quite marginal (especially for MB-453). The inflexion point of the curves appears approximately equal for all concentrations of RIDR-PI-103.

                3.5 – as per Figure 1, please relabel the x-axis scale.

(4) Sections 2.3, 2.4 and 2.5 – the data does not support the notion that doxorubicin sensitises cells to RIDR-PI-103. As both compounds are active, the argument could be made that RIDR-PI-103 sensitises cells to doxorubicin. From the data, it appears that the combination of RIDR-PI-103 + dox is additive compared to either agent alone.

(5) Figure 6 – the original images are absent from the SI as per the journal instructions to authors “In order to ensure the integrity and scientific validity of blots (including but not limited to Western blots) and gel data reporting, original, uncropped and unadjusted images should be uploaded as Supporting Information files at time of initial submission.”

(7) In the PK study, was 20 mg/kg the maximum solubility of RIDR-PI-103 in the vehicle used? Please note in the text if it was. I assume that RIDR-PI-103 was therefore dosed as a solution and not a suspension. It would be useful to clarify this in the text.

(8) Figure 7 – given that RIDR-PI-103 is a pro-drug, were the levels of PI-103 in the plasma samples also measured? This is almost as important to understand as the levels of the pro-drug.

(9) Section 4.10 Animal Studies:

                9.1 – was ethical approval for the animal studies obtained and from whom? To what standards did the animal unit work to etc?

                9.2 – line 424 – the authors state serial blood collections were taken. This implies repeated sampling from the same animal but they also state n=3 mice per time point. Please correct. How was the blood collected? Cardiac puncture under anaesthetic?

(10) I struggled with this paper to understand the benefit of RIDR-PI-103 in combination with dox over an established PI3K inhibitor (e.g. Pictilosib etc) in combination with dox. What are the perceived advantages? I would expect there to be no difference in tox (especially cardiotox) as dox will lead to increased ROS in any tissues it is distributed to and therefore convert RIDR-PI-103 to PI-103 in those tissues also.

Minor points

(11) The standard nomenclature for Checkpoint kinase 1 (or 2) is either Chk1 or CHK1 not ChK1. Can these be amended.

(12) Line 317 – RPM1 should be RPMI

(13) Section 4.1 – what solvent were the compounds prepared in? DMSO?

(14) Lines 344 and 348 – doses implies administration to an animal. Concentrations might be more appropriate.

(15) Line 350 – was the MB-361 concentration range of Docetaxel really 10-90 nM? Given the other 2 cell lines had a 1000 and 200-fold difference between min and max concentrations, this seems a very small range.

(16) Section 4.6 – what instrument was used to analyse DCFH-DA levels? I’m assuming a BD Flow Cytometer based on the software mentioned.

(17) In the abstract (line 29), it would be useful to give the concentration (μM) of RIDR-PI-103 that equates to 200 ng/mL.

Reviewer 2 Report

This study evaluated the in vitro effects of RIDR-PI-103 on breast cancer cells (MDA-MB-231, MDA-MB-361 and MDA-MB-453), non-tumorigenic MCF10A and fibroblasts. The effects of RIDR-PI-103 combined with doxorubicin on Matrigel colony formation, cell proliferation and migration. Pharmacokinetic (PK) studies using C57BL/6J mice determined systemic exposure (plasma concentrations and overall Area Under the Curve) and T1/2 of RIDR-PI-103.

The authors claimed that MDA-MB-453, MDA-MB-231 and MDA-MB-361 cells were sensitive to RIDR-PI-103 vs MCF10A and normal fibroblast. Doxorubicin induced ROS sensitized RIDR-PI-103 to suppress cancer cell growth and proliferation. Doxorubicin with RIDR-PI-103 inhibited p-AktS473, upregulated p-Chk1/2 and p-P53. PK studies showed that ~200 ng/ml RIDR-PI-103 is achievable in mice plasma with an initial dose of 20 mg/kg and a 10 hr T1/2. They concluded that RIDR-PI-103 prodrug sensitized by doxorubicin could be a potential therapeutic for treatment of breast cancer patients.

However, several defects are detected and the following concerns should be addressed.

  1. The authors claimed that doxorubicin induces ROS to activate RIDR-PI-103 to exhibit the cytotoxic effects on breast cancer cells. However, no any direct evidence was provided to support the “ROS” induced by doxorubicin contributes to the observed cytotoxic effects. The reviewer suggests that the authors should determine whether any antioxidant treatment could prevent the activation of RIDR-PI-103 by doxorubicin.
  2. Figure I and Figure 2 show that the endogenous ROS levels at untreated and treated status by doxorubicin are not consistent with the cytotoxic effects of RIDR-PI-103 and doxorubicin. The authors should explain the reasons.
  3. In Figure 1, no PI-103 data were shown. The authors should provide these data.
  4. For transwell migration analysis, the authors assessed the migrated cells after 24-48 hours. The migration time is longer than the cell doubling time, which might be affected by cell proliferation. The reviewer suggest that the authors should analyze the migrated cells within 6-12 hours for cell migration.
  5. In Abstract, the authors mentioned that they examined the invasive and migratory properties. However, no any cell invasion data were shown. The data should be provided.
  6. In this study, the authors studied Pharmacokinetic (PK) using C57BL/6J mice determined systemic exposure of RIDR-PI-103. However, the in vivo anticancer effects and cardiotoxicity of RIDR-PI-103 combined with doxorubicin are more important and should be addressed.

Round 2

Reviewer 1 Report

I thank the authors for the changes made to the manuscript and the extra data supplied, I feel this has improved the manuscript. It is a shame the authors have not yet measured conversion of RIDR-PI-103 to PI-103 in the mouse plasma as this would be really interesting to see.

There is one minor (apologies) correction that requires to be made to figure legend 2A - the authors need to define what *, ** and # represent. I am assuming these are p values but it would be good to have these defined.

Author Response

Please see attached letter.

Reviewer 2 Report

The authors addressed most of the reviewer’s concerns and made revisions in the revised manuscript. However, the major issue for the ROS role in this study are still not addressed. The reviewer suggests that the authors should determine whether any antioxidant treatment could prevent the activation of RIDR-PI-103 by doxorubicin.

Author Response

Please see attached letter.

Round 3

Reviewer 2 Report

The revised version is improved.